# Experience of Combined Procedure during Percutaneous LAA Closure

**DOI:** 10.3390/jcm11123280

**Published:** 2022-06-08

**Authors:** Guillaume Domain, Nicolas Dognin, Gilles O’Hara, Josep Rodès-Cabau, Jean-Michel Paradis, Camille Strubé, Mathieu Bernier, Kim O’Connor, Jonathan Beaudoin, François Philippon, Erwan Salaun, Jean Champagne

**Affiliations:** Institut Universitaire de Cardiologie et de Pneumologie de Québec, Quebec City, QC G1V 4G5, Canada; nicolas.dognin@gmail.com (N.D.); gilles.ohara@fmed.ulaval.ca (G.O.); josep.rodes@criucpq.ulaval.ca (J.R.-C.); jean-michel.paradis@criucpq.ulaval.ca (J.-M.P.); camille.strube@gmail.com (C.S.); mathieu.bernier@criucpq.ulaval.ca (M.B.); kim.oconnor@criucpq.ulaval.ca (K.O.); jonathan.beaudoin@criucpq.ulaval.ca (J.B.); francois.philippon@fmed.ulaval.ca (F.P.); erwan.salaun.1@ulaval.ca (E.S.); jean.champagne@fmed.ulaval.ca (J.C.)

**Keywords:** left atrial appendage closure, combined procedures, atrial flutter ablation, watchman device, atrial fibrillation, stroke

## Abstract

Introduction: Percutaneous left atrial appendage closure (LAAC) is an alternative to oral anticoagulants (OAC) in patients with non-valvular atrial fibrillation (AF) and contraindication to long-term OAC. Combined strategy with percutaneous LAAC at the same time of other cardiac structural or electrophysiological procedures has emerged as an alternative to a staged strategy. Aim: To describe our experience with combined LAAC procedures using Watchman™ devices. Methods: All patients with combined LAAC procedures using Watchman™ (WN) devices performed from 2016 to 2021 were included. The primary safety endpoint was a composite of periprocedural complications and adverse events during the follow-up. The primary efficacy endpoint included strokes, systemic embolisms, major bleeding and cardiovascular death. Results: From 2016, among 160 patients who underwent LAAC using WN devices, 19 underwent a combined strategy: 7 transcatheter edge-to-edge mitral valve repair (TEMVR) (37%), 6 typical atrial flutter ablation (31%), 2 leadless pacemaker (LP) implantation (10%) and 4 AF ablation (22%). The WN device was successfully implanted in 98% and 100% of cases for single and combined LAAC procedures, respectively (*p* = 0.63). Median follow-up was 13 months (IQR 25/75 3/24). Device-related complications occurred in 6 out of 141 patients (4%) who underwent single LAAC and in no (0/19) patient in the combined LAAC procedure (*p* = ns). The procedural-related complications did not differ significantly between groups (5% vs. 10%, respectively, in the single and combined group, *p* = 0.1). Conclusion: Combined procedure using the Watchman™ devices and one other structural or electrophysiological procedure appears safe and effective. Larger series are needed to confirm these results.

## 1. Introduction

Oral anticoagulation (OAC) is the cornerstone of stroke prevention in non-valvular atrial fibrillation (NVAF) according to the CHADS-VASC score [1,2]. However, comorbidities, personal history of bleeding or persistent risk of bleeding remains a frequent contraindication for long-term OAC [3]. In NVAF, thrombi typically form in the left atrial appendage [4]. Over the last years, LAAC with an occluder device has emerged as an alternative to OAC in selected patients [5,6].

Different design and generations of devices are available for percutaneous LAA closure: Watchman™ (Boston Scientific, St. Paul, MN, USA), ACP™ (Abbott, Chicago, IL, USA), Amulet™ (Abbott, Chicago, IL, USA), WaveCrest™ (Biosense Webster, Irvine, CA, USA), and LAmbre™ (Life Tech Sci, Shenzhen, China) devices [7].

Recently, a combined strategy of concomitant LAAC for stroke prevention and catheter ablation (CA) for AF was proposed, and an international multicenter registry supports the feasibility and safety of this strategy [8,9]. The combined strategy of concomitant CA and LAAC in symptomatic AF patients with high risk of stroke and bleeding may emerge as a cost-effective therapeutic option compared to CA and long-term oral anticoagulation (OAC) [10]. Moreover, in patients with both NVAF and a patent foramen oval (PFO) or an atrial septal defect (ASD), LAAC combined with PFO or ASD closure has been previously reported [11,12]. Finally, other percutaneous procedures such as leadless intracardiac pacing systems implantation or transcatheter edge-to-edge mitral valve repair (TEMVR) may be combined with LAAC [13,14,15,16]. Recent publications have shown the feasibility of LAAC using Amplatzer^TM^ devices combined with structural, coronary, or electrophysiological procedures [16]. A combined approach could allow treatment of several cardiac conditions in a single intervention. Most of the recent publications have focused on a combination of CA for AF and LAAC [8].

Here we report our single-center experience looking at the efficacy and safety of a combined strategy with LAAC using WN devices and other percutaneous procedures.

## 2. Methods

This study was conducted according to ethical standards of clinical e-research in Canada and in accordance with the declaration of Helsinki. This is a retrospective analysis of clinical, biological, and echocardiographic data prospectively collected in a single-center registry of all patients (n = 160) who underwent percutaneous LAAC using WN devices at our institution from October 2016 to October 2021. A combined strategy was defined as a LAAC closure with WN devices associated with concomitant cardiac structural or electrophysiological interventions using the same femoral venous access.

### 2.1. Definitions and Outcomes Measures

Immediate and delayed procedural-related complications were collected according to the Munich Consensus Document [17].

Procedural success was defined as: 1-technical success and 2-no procedure-related complications.

Technical success was defined as: (1) exclusion of the LAA; (2) no device-related complication; and (3) no leak ≥ 5 mm on color Doppler TEE.

Device-related complications included device-related thrombus (DRT), device embolization, erosion, interference with the surrounding structure (circumflex coronary artery, mitral valve, pulmonary artery, or pulmonary vein), fracture, perforation or laceration, infection, or endocarditis.

Based on the Protect AF trial, an adequate sealing of the LAA was defined as a jet < 5 mm. A jet ≥ 5 mm was considered as a significant para-prosthetic leak [18].

The procedural-related complications included stroke (hemorrhage or infarction), transient ischemic attack, systemic embolism, life-threatening or major bleeding, pericardial effusion, vascular complications, pericarditis, myocardial infarction, renal failure, hepatic failure, cardiovascular death, and unknown cause of death during the follow-up. Major bleeding was defined as one of the following criteria: (1) fatal bleeding; (2) symptomatic bleeding in a critical organ (intracranial, intraspinal, intraocular, retroperitoneal, intraarticular, pericardial, intramuscular with compartment syndrome); (3) a fall in the hemoglobin level of ≥20 g/L; (4) transfusion of two or more units of whole blood or red cells [19,20]. Ischemic stroke was defined as an episode of neurological dysfunction caused by a focal cerebral, spinal, or retinal infarction and could be definitive, transient, or silent [21]. Data related to peripheral embolism and hemorrhagic strokes were also collected. The CHA_2_DS_2_-VASc and HASBLED scores were calculated.

### 2.2. Procedures

The LAAC devices used in this study were the Watchman 2.5^TM^ and the Watchman FLX^TM^ devices (Boston Scientific, St. Paul, MN, USA) using the TruSeal access sheath (Boston Scientific, St. Paul, MN, USA). The basic steps of the procedure were similar for the Watchman 2.5^TM^ and FLX^TM^ devices regarding the use of general anesthesia, femoral venous access for the transseptal puncture, use of a guide wire and pigtail for sheath guidance and positioning, and device selection based on transesophageal echocardiography (TEE) and angiography. Implantation of the Watchman 2.5^TM^ devices was performed as recommended [22], and the ball technique was used for the implantation of the Watchman FLX devices [22]. After satisfactory TEE assessment of the standard PASS criteria (position, anchor, size/compression, seal), fluoroscopic morphology and angiographic test, a tug test was performed. The Watchman device was then released, and all material removed. Patients could receive a protamine infusion and/or venous closure devices, or hemostatic suture and a compressive dressing.

### 2.3. Postoperative Care

After the procedure, all patients stayed in the cardiac care unit for 24 h continuous ECG monitoring, and transthoracic echocardiography (TTE) and chest X-ray were performed before discharge. For the following 45 to 60 post-operative days, all patients received either oral anticoagulation, dual or single antiplatelets according to their clinical profile. The antithrombotic regimen was then prescribed taking into consideration any residual leak, any device-related thrombus or adverse outcome.

### 2.4. Follow-Up

Patients were followed in our specialized LAA clinic at 6 weeks, 6 months, 1 year, and 2 years after their procedure. During the follow-up visits, clinical status and ECG were recorded. A TEE was performed at 45–60 days post procedure, then repeated at 1-year follow-up to assess any residual leak and PASS criteria and to exclude any complication. If a TEE was contraindicated, a standard TTE was performed.

### 2.5. Statistical Analysis

Continuous data were expressed as median and interquartile range (IQR) or mean and standard deviation (SD) and compared between groups using ANOVA. Qualitative variables were presented as percentages and compared between groups using Chi^2^ test or Fisher’s exact test. A univariate Cox model was performed to compare the occurrence of events in each group.

All tests were two-sided. A *p*-value < 0.05 was considered significant for all analysis.

R Studio™ statistical software (RStudio Inc., Boston, MA, USA 2019 version, 1.2.5001) was used.

## 3. Result

### 3.1. Patient Demographics

From 160 patients who underwent LAAC using WN devices, 19 (12%) underwent a combined strategy. Overall, the median follow-up was 13 months (IQR 25/75 3/24). The median follow-up was 13 months (IQR 25/75 4/24) for the single group and 7 months (IQR 25/75 1/24) for the combined group (*p* = 0.1). The median age was 76 years (IQR 25/75 71/80) and 71 years (IQR 25/75 61/73) in the single and combined strategy groups, respectively (*p* < 0.01). The CHA2DS2-VASc score (mean ± SD) was 4 (IQR 25/75 3/5) in each group, whereas the HAS-BLED score was 4 ± 1 in the single and 3 ± 1 in the combined strategy (*p* = 0.03). Atri^3^al fibrillation was permanent in 47% of patients in the single and in 37% in the combined strategy (*p* = ns). Indications for LAAC are depicted in Table 1.

### 3.2. Device Implantation Outcomes

The LAAC immediate device implantation success rate was not significantly different (*p* = 0.63) between groups, 98% in the single procedure group and 100% in the combined strategy group. Two procedural failures were related to unfavorable anatomy and a high risk of prosthesis embolization after assessment of the PASS criteria. The median procedural time, fluoroscopy time and fluoroscopy dose were significantly higher in the combined strategy group (*p* < 0.01). Recapture or device resizing was required in 42 patients (27%) (Table 2).

The combined strategy included 7 TEMVR, 6 typical atrial flutter ablations (AFL), 2 leadless pacemaker (LP) implantations and 4 atrial fibrillation ablations. Among the 6 patients undergoing a typical flutter ablation, 3 were in atrial flutter at time of LAAC and sinus rhythm was obtained in all. Procedural time, fluoroscopy time and fluoroscopy dose were significantly lower for combined procedures with AFL ablation (Table 3).

### 3.3. Mid-Term Device-Related Complications

Six patients in the single procedure group had a device-related complication during follow-up: Three had a significant para-prosthetic leak (with a jet size > 5 mm), and 3 a DRT. No device embolization occurred during follow-up in the whole cohort (Table 3). Thus, the mid-term overall success rate was 96% and 100% in the single procedure group and in the combined strategy group, respectively (*p* = ns).

### 3.4. Mid-Term Procedural-Related Complications

Procedural-related complications occurred in 8 and 2 patients in the single procedure group and in the combined strategy group, respectively (*p* = ns). One patient in each group presented with a major hemorrhage due to recurrent gastrointestinal bleeding on Aspirin (at 2 years in the single group and 7 months in the combined strategy with TEMVR (*p* = ns)). Cardiovascular death occurred in 3 patients in the single procedure group due to hemorrhagic stroke in 2 and end-stage heart failure in 1. One death in the combined group was caused by bladder neoplasia (Table 4).

### 3.5. Staged Procedures

Among the single procedure group, several patients (n = 66, 42%) had a structural or electrophysiological intervention before or after the LAAC. Twenty patients underwent atrial flutter ablation (13%), 7 CA for atrial fibrillation ablation (4%), 43 (27%) received a pacemaker (17 VVI, 19 DDD and 7 CRT) and TEMVR was performed in one patient. These staged interventions were performed less than 12 months before or after LAAC in 27 patients (17%).

## 4. Discussion

The key findings of our study are that a combined strategy with LAAC using Watchman™ devices and one other cardiac structural or electrophysiological interventions using the same venous femoral access: (1) is not associated with a lower procedural success and (2) remains safe and effective at mid-term follow-up (Figure 1).

### 4.1. Multiples Interventions and Risk in the Elderly Population

In the frail and elderly population referred for LAAC, patients harbor several cardiac and extra-cardiac comorbidities [23]. This clinical status leads to both: (1) a higher risk of requiring multiple cardiac interventions; and (2) a higher risk of complications when these cardiac interventions are performed. Moreover, multiple and repeated hospitalizations and anesthesia/conscious sedation in the elderly population can result in periprocedural complications unrelated to the initial clinical condition that prompted the admission or to the procedure itself [24]. Thus, a strategy combining LAAC and another cardiac structural or electrophysiological intervention in a single intervention appears attractive. In fact, the combined strategy may decrease hospitalizations and the length of stay and would require only one anesthesia/sedation. On the other hand, due to comorbidities and age, patients may be at higher risk of periprocedural morbidity and mortality [25,26]. Thus, the combined strategy has to be studied to demonstrate equal or even superior benefits in terms of efficacy and safety. In our small cohort, despite similar baseline characteristics, the rates of technical and procedural success and periprocedural complications were not different in patients who underwent the combined strategy compared to those who underwent the single LAAC. These results strengthen the feasibility and safety of the concomitant approach previously described in the Swiss series using the Amplatzer^TM^ devices [16].

### 4.2. Type of Interventions and Procedural Issues

Half of the patients referred for LAAC in our center had a second cardiac intervention either in a combined strategy or during a second procedure. These second interventions, whether structural or electrophysiological, were also performed from a venous femoral access. Thus, the combined strategy using the same venous femoral access can reduce the cumulative risk of vascular complications and length of stay. Moreover, the same transeptal puncture when required for AF ablation or TEMVR may be used for the LAAC [16]. However, the optimal transeptal position may be different for TEMVR and LAAC, and the use of the optimal TEMVR transeptal puncture position should be preferred and used for the LAAC. In our experience, when AF ablation or TEMVR are combined, LAAC is performed last. However, when the other intervention is a flutter ablation or leadless pacemaker implantation, LAAC is performed first under general anesthesia.

LAAC can be combined with other interventions requiring an arterial access. The transaortic valve replacement (TAVR) is probably the procedure that could lead to such a combined strategy [16]. In this approach, TAVR is first performed from an arterial access and LAAC, then performed from a venous femoral access.

However, more procedures are now done using conscious sedation. Most TAVR, AF or flutter ablations, and LP implantations are performed without general anesthesia, while LAAC and TEMVR often require TEE guidance and general anesthesia [27,28]. The use of intracardiac echocardiography may obviate the need for general anesthesia [29].

### 4.3. Perspectives

With the development of percutaneous interventions and therapies, the use of combined strategies will increase. This approach may reduce the overall cost for the healthcare system reducing hospitalizations and the need for staged interventions [30,31]. Moreover, the Heart Team approach will increase collaboration between different subspecialties such as the electrophysiologists, geriatricians, anesthesiologists, echocardiographists and the structural specialists all oriented towards better patient care and outcome. With this approach, in high volume centers, we may expect a decrease in the complication rates, more efficiency and a decrease in costs [31,32].

### 4.4. Study Limitations and Future Directions

Multiple limitations arise from a monocentric registry design including variation in implantation modality, post-discharge anticoagulation regimen and the difficulty to extrapolate to other centers or countries.

Our sample size is small. Follow-up times differed between the 2 groups. These limitations must be considered in interpretation of the results, especially for the procedural rate complications, which was not significantly different between the two groups.

Furthermore, since the registry dataset was primarily focused on LAAC results, other data were not prospectively collected (arrhythmia, rhythm at follow-up, or valvular outcomes) for the combined procedure and were assessed by chart review.

Additionally, since no independent image adjudication was used, all TEE measurements (LAA diameter, device size, compression, peri-device leak, and device thrombosis) are subject to operator interpretation and imaging system variability.

## 5. Conclusions

A combined procedure using LAAC with the Watchman ^TM^ devices, and another cardiac structural or electrophysiological procedure, appears safe and effective. Larger series and prospective and multicentric cohorts are needed to confirm these preliminary results. However, since many patients have a clinical indication for multiple cardiac structural or electrophysiological procedures (TEMVR, atrial flutter ablation, pacemaker implantation), the combined approach may be considered.

## Figures and Tables

**Figure 1 jcm-11-03280-f001:**
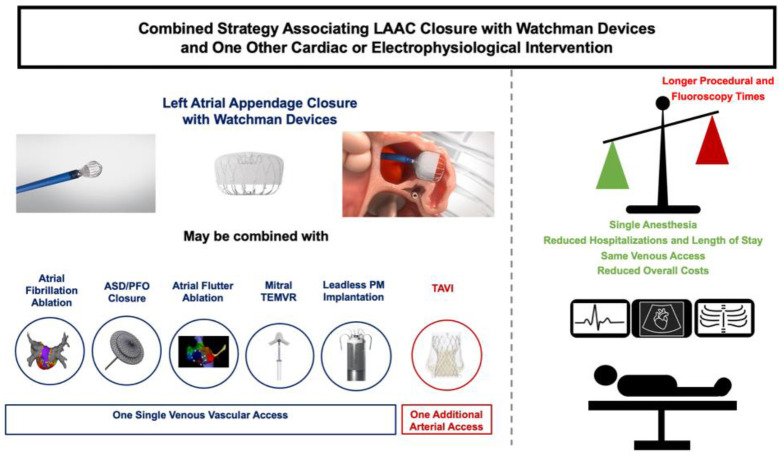
Benefits and limitations of the combined strategy during LAAC. Image provided courtesy of Boston Scientific. ©2021 Boston Scientific Corporation or its affiliates. All rights reserved.

**Table 1 jcm-11-03280-t001:** Baseline characteristics.

	Single Procedure (n = 141)	Combined Strategy (n = 19)	*p*-Value
Age (years)	76 (71/80)	71 (61/73)	<0.01
Male	94 (66%)	10 (66%)	0.73
Hypertension	118 (84%)	14 (87%)	0.69
Diabetes	52 (37%)	8 (50%)	0.31
Dyslipidemia	59 (42%)	11 (69%)	0.05
History of stroke	35 (25%)	3 (19%)	0.59
LVEF (%)	53 (50/60)	45 (37/56)	<0.01
Left atrium volume (mL/m^2^)	44 (35/52)	48 (37/61)	0.23
Coronary Heart Disease	66 (47%)	10 (62%)	0.23
Valvular Heart Disease	64 (45%)	7 (43%)	0.90
Abnormal renal function	39 (27%)	4 (25%)	0.82
Abnormal liver function	9 (7%)	0 (0%)	0
COPD	18 (13%)	3 (19%)	0.50
Peripheral artery disease	4 (3%)	0 (0%)	0
History of major bleeding	127 (90%)	12 (75%)	0.07
Gastrointestinal hemorrhage	63 (49%)	9 (56%)	0.38
Intracerebral bleeding	33 (26%)	1 (6%)	0.22
Hematuria	12 (9%)	1 (6%)	0.75
Others	19 (15%)	1 (6%)	0.81
Blood Dyscrasia	12 (8%)	1 (6%)	0.24
Refractory anemia	37 (26%)	5 (31%)	0.31
Combined procedure	0 (0%)	19 (100%)	0
Atrial flutter ablation	0 (0%)	6 (31%)	0
Leadless pacemaker implantation	0 (0%)	2 (10%)	0
TEMVR	0 (0%)	7 (37%)	0
Atrial fibrillation ablation	0 (0%)	4 (22%)	0

Continuous data were expressed as median and IQR (25/75). Qualitative variables were presented with number and percentages. Abbreviations: AF = Atrial Fibrillation; CHA_2_DS_2_-VASc score = congestive heart failure, hypertension, 75 years of age and older, diabetes mellitus, previous stroke or transient ischemic attack, vascular disease, 65 to 74 years of age, female; HAS-BLED score = hypertension, abnormal renal/liver function, stroke, bleeding history or predisposition, labile international normalized ratio, elderly, drugs/alcohol concomitantly; LVEF = Left ventricular ejection fraction; COPD = chronic obstructive pulmonary disease, TEMVR = Transcatheter Edge-to-Edge mitral valve Repair.

**Table 2 jcm-11-03280-t002:** Left Atrial Appendage Closure Procedures.

	Single Procedure (n = 141)	Combined Strategy (n = 19)	*p*-Value
LAA ostial diameter (mm)	19 (17/21)	20 (18/21)	0.46
Number of device deployments	1 (1/2)	1 (1/1)	0.43
Number of devices	1 (1/1)	1 (1/1)	0.86
Device compression (%)	20 (15/23)	18 (15/20)	0.49
Procedural time (min)	69 (53/88)	115 (76/139)	<0.01
Fluoroscopy time (min)	7 (5/10)	18 (10/38)	<0.01
Fluoroscopy dose (mGy)	945 (402/2760)	2500 (640/11,992)	<0.01
Success	139 (98%)	16 (100%)	0.63

Continuous data were expressed as median and IQR (25/75). Qualitative variables were presented with number and percentages. Abbreviations: LAA = left atrial appendage.

**Table 3 jcm-11-03280-t003:** Atrial Flutter Ablation Procedures.

	Atrial Flutter Ablation (n = 6)	Other Combined Procedures (n = 13)	*p*-Value
LAA ostial diameter (mm)	20 (18/21)	19 (18/21)	0.55
Number of device deployment	1 (1/1)	1 (1/2)	0.18
Number of devices	1 (1/1)	1 (1/1)	0.46
Device compression (%)	17 (15/19)	20 (15/21)	0.56
Procedure time (min)	75 (64/80)	133 (123/148)	<0.01
Fluoroscopy time (min)	8 (4/11)	36 (22/56)	<0.01
Fluoroscopy dose (mGy)	512 (336/769)	7820 (3134/15,300)	0.01

Continuous data were expressed as median and IQR (25/75). Qualitative variables were presented with number and percentages. Abbreviations: LAA = left atrial appendage.

**Table 4 jcm-11-03280-t004:** Midterm Outcomes in Single LAAC Procedures and Combined Strategy.

	Single LAAC (n = 141)		Combined Strategy (n = 19)			
	Number of Patients	Rate (%)	Number of Patients	Rate (%)	Hazard Ratio [95% CI]	*p*-Value
Device-related complications	6	4	0	0	0	0
Thrombosis	3	2	0	0	0	0
Device embolization	0	0	0	0	0	0
Leak (>5 mm)	3	2	0	0	0	0
Technical success	135	96	19	100	1.1 [0.6–2]	0.6
Procedural-related complications	8	5	2	10	3.5 [0.7–17]	0.1
Ischemic stroke	1	1	0	0	0	0
SE	0	0	0	0	0	0
Hemorrhagic stroke	3	2	1	5	5.6 [0.5–6]	0.2
Bleeding	1	1	1	5	12 [0.8–201]	0.07
CV/unknown death	3	2	0	0	0	0
Pericardial effusion	2	1	0	0	0	0
Vascular complications	0	0	0	0	0	0
Pericarditis	0	0	0	0	0	0
Procedural success	129	91	17	89	1 [0.7–2.2]	0.4
All death	8	5	2	12	3,6 [0.7–17]	0.1

Qualitative variables were presented with number and percentages. Abbreviations: SE = systemic embolism; CV = cardiovascular.

## Data Availability

Data available on request due to restrictions e.g., privacy or ethical.

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
