# Peer review of "Experience of Combined Procedure during Percutaneous LAA Closure"

_jcm, 2022, doi:10.3390/jcm11123280_

Round 1

Reviewer 1 Report

In the present resubmitted manuscript, Domain and co-workers aimed to report the single centre experience of left atrial appendage closure (LAAC) combined with  other percutaneous procedures. However in fact the authors assess the efficacy and safety of LAAC procedure between the groups where  (1) LAAC was performed exclusively  or (2) on the top of LAAC typical atrial flutter ablation, atrial fibrillation ablation, leadless pacemaker implantation  or transcatheter mitral valve repair were applied. In the presented results besides some obvious parameters such as longer procedural time or fluoroscopy time during a combined procedure no differences were noted between the groups  The authors concluded that a combined procedure with LAAC appears to be safe and effective.

Major issues

1.       A p-value of 0.05 or lower is generally considered statistically significant. The authors considered a p-value < 0.5 as significant for all analysis (line 126). It seems that analysed dataset was completely wrongly assessed.

2.       The authors provide the different numbers of analysed study population throughout the manuscript. It is written that the study cohort consisted of  157 patients (line 62), whereas 160 patients underwent LAAC (line 16, line 130, all tables). Moreover  it is written that 19 patients underwent a combined procedure (line 16, line 130, all tables) whereas within  the table 1 only 16 patients underwent this procedure.

3.       Comparing  the efficacy and safety of LAAC procedure  one does not expect any procedural differences when it is performed as a standalone procedure or the other procedure is added  on the top of LAAC. Only efficacy and safety of the other procedure matters. Therefore the weakness of study is the lack of novelty.

4.       The number of patient who underwent a combined procedure  is small  (16 or 19 patients?), follow-up is very short (5 months), and  on the top of LAAC many different procedures were applied. 

5.       All tables provide results that are  completely  statistically insignificant. Moreover all results are repeated in the main text.

6.       Many different procedures were applied on the top of LAAC  that have different impact on the possible complications of combined strategy, e.g.    typical atrial flutter ablation is a quite short-lasting and very safe procedure whereas atrial fibrillation ablation is usually long-lasting procedure and  requires constant anticoagulation. Therefore the procedural risk is much higher. Moreover the authors do not provide any details of combined strategy.

Author Response

We thank the reviewer for his review and valuable comments to improve our article.

“1. A p-value of 0.05 or lower is generally considered statistically significant. The authors considered a p-value < 0.5 as significant for all analysis (line 126). It seems that analysed dataset was completely wrongly assessed.”

  • We thank the reviewer for his comment. This is a typing error. The p-value considered significant is 0.05. The manuscript has been corrected.

“2. The authors provide the different numbers of analysed study population throughout the manuscript. It is written that the study cohort consisted of 157 patients (line 62), whereas 160 patients underwent LAAC (line 16, line 130, all tables). Moreover it is written that 19 patients underwent a combined procedure (line 16, line 130, all tables) whereas within the table 1 only 16 patients underwent this procedure.”

  • We thank the reviewer for his comment. This is a mistake. The manuscript has been corrected.

“3. Comparing the efficacy and safety of LAAC procedure one does not expect any procedural differences when it is performed as a standalone procedure or the other procedure is added on the top of LAAC. Only efficacy and safety of the other procedure matters. Therefore the weakness of study is the lack of novelty.”

  • We thank the reviewer for his comment. We agree that the effectiveness and safety of LAAC or the associated procedure should not be different when performed separately or simultaneously. This is what we want to raise in this work.

“4. The number of patient who underwent a combined procedure is small (16 or 19 patients?), follow-up is very short (5 months), and on the top of LAAC many different procedures were applied.”

  • This is indeed a single-center, real-life study. The small number of patients and the limited follow-up give it a low level of scientific proof. We have mentioned this element in the discussion. However, we find this work relevant in the sense that it describes the feasibility of combined procedures in current practice with different interventions. Notably, atrial flutter ablation associated with LAAC has never been documented in the literature.

“5. All tables provide results that are completely statistically insignificant. Moreover all results are repeated in the main text.”

  • The lack of statistical power due to the small number of participants explains the non-significant results. The most notable analyses were those concerning procedure time, duration, and fluoroscopy dose.

“6. Many different procedures were applied on the top of LAAC that have different impact on the possible complications of combined strategy, e.g.    typical atrial flutter ablation is a quite short-lasting and very safe procedure whereas atrial fibrillation ablation is usually long-lasting procedure and requires constant anticoagulation. Therefore, the procedural risk is much higher. Moreover, the authors do not provide any details of combined strategy.”

  • We thank the reviewer for this comment. The complexity of the associated procedures is indeed different. We consider it an important point that the performance of a simple or complex procedure associated with LAAC does not alter the effectiveness and safety of the LAAC procedure. All the procedures associated with LAAC have been performed according to the current guidelines. We have chosen not to describe them so as not to overshadow the message of our work. The specifics of the transseptal procedure have been described in the discussion.

Reviewer 2 Report

The current paper's version has several improvements. My previous comments have been answered, so I have no other relevant issues.

Author Response

We thank the reviewer for his review.

Reviewer 3 Report

The study is an interesting single-center experience of combined procedure during left atrial appendage closure.

Since the percentage of complications in the combined strategy group is not low (probably related to small sample size), larger series are needed to confirm these results.

Author Response

We thank the reviewer for his review. 

Round 2

Reviewer 1 Report

In the revised version of manuscript the authors have corrected all indicated  typing errors. However there are still several caveats that must be corrected.

    1. Responding to my comments the authors concluded that:

a        a)     They find their work relevant as it describes the feasibility of combined procedures in the current practice with different interventions

b)    They want to raise the fact that the effectiveness and safety of  LAAC using a Watchman device  as well as an associated procedure is not different when performed separately or simultaneously.

c)     It is the first report concerning above-mentioned issues in patients who underwent atrial flutter on the top of  LAAC

As far as I am concerned the information that the authors aimed  to assess the feasibility and safety of combined procedures should be clearly stated in the introduction.

2. Results are repeated  in the main text and tables. The information should  not be doubled. If the authors want to provide the information in the main text it should be deleted from the tables or vice-versa.  

3. The text should be carefully checked to detect syntax errors  e.g. an international multicenter registry supports (not support) the feasibility and safety of this strategy (line 42)

Author Response

We thank the reviewer for his review and comments to improve our paper.   "1. As far as I am concerned the information that the authors aimed to assess the feasibility and safety of combined procedures should be clearly stated in the introduction."

  • We thank the reviewer for his comment. We add a sentence in the manuscript, at the end of the introduction. 

"2. Results are repeated  in the main text and tables. The information should  not be doubled. If the authors want to provide the information in the main text it should be deleted from the tables or vice-versa."

  • We thank the reviewer for his comment. The manuscript has been simplified and duplicate results removed. 

"3. The text should be carefully checked to detect syntax errors  e.g. an international multicenter registry supports (not support) the feasibility and safety of this strategy (line 42)."

  • We thank the reviewer for his comment. The manuscript has been corrected. The English language and syntax have been completely revised.